# Understanding of arthrofibrosis: New explorative insights into extracellular matrix remodeling of synovial fibroblasts

**Thanh-Diep Ly[1], Meike Sambale[1], Lara Klösener[1], Philipp Traut[2], Bastian Fischer[1], Doris Hendig[1], Joachim Kuhn[1], Cornelius Knabbe[1], Isabel Faust-Hinse[1] ***

1 Institut für Laboratoriums- und Transfusionsmedizin, Herz- und Diabeteszentrum NRW, Universitätsklinik der Ruhr-Universität Bochum, Bad Oeynhausen, North Rhine-Westphalia, Germany, 2 Orthopädische Beratung und Begutachtung, Bad Oeynhausen, North Rhine-Westphalia, Germany

* ifaust-hinse@hdz-nrw.de

**Data Availability Statement:** All relevant data are within the paper and its Supporting information files.

## Abstract

Arthrofibrosis following total knee arthroplasty is a fibroproliferative joint disorder marked by dysregulated biosynthesis of extracellular matrix proteins, such as collagens and proteoglycans. The underlying cellular events remain incompletely understood. Myofibroblasts are highly contractile matrix-producing cells characterized by increased alpha-smooth muscle actin expression and xylosyltransferase-I (XT-I) secretion. Human XT-I has been identified as a key mediator of arthrofibrotic remodeling. Primary fibroblasts from patients with arthrofibrosis provide a useful *in vitro* model to identify and characterize disease regulators and potential therapeutic targets. This study aims at characterizing primary synovial fibroblasts from arthrofibrotic tissues (AFib) regarding their molecular and cellular phenotype by utilizing myofibroblast cell culture models. Compared to synovial control fibroblasts (CF), AFib are marked by enhanced cell contractility and a higher XT secretion rate, demonstrating an increased fibroblast-to-myofibroblast transition rate during arthrofibrosis. Histochemical assays and quantitative gene expression analysis confirmed higher collagen and proteoglycan expression and accumulation in AFib compared to CF. Furthermore, fibrosis-based gene expression profiling identified novel modifier genes in the context of arthrofibrosis remodeling. In summary, this study revealed a unique profibrotic phenotype in AFib that resembles some traits of other fibroproliferative diseases and can be used for the future development of therapeutic interventions.

## Introduction

Arthrofibrosis is a pathological condition characterized by excessive scar tissue formation in a joint resulting in joint stiffness and a reduction in the normal range of joint motion. There is no cure for arthrofibrosis available currently, but clinicians use nonsurgical and surgical procedures to diminish it. Arthrofibrosis can occur in most joints because it is a disorder of the synovial membrane, but the knee is the most common site where arthrofibrosis is likely to follow after an orthopedic procedure, such as total knee arthroplasty (TKA) [1–3]. Arthrofibrosis has been shown to account for 10% of TKA revision (RTKA-A) that are performed

**Funding:** The authors received no specific funding for this work.

**Competing interests:** The authors have declared that no competing interests exist.

within five years after primary TKA surgery. Histopathological findings and clinical symptoms, such as stiffness and limited range of joint motion, are commonly used for the diagnosis of arthrofibrosis [4]. The pathology of arthrofibrosis following TKA is incompletely understood but involves the dysregulation of physiological wound healing responses [5]. On the cellular level, arthrofibrosis is caused by an increased myofibroblast proliferation and reduced apoptosis rate, leading to a higher myofibroblast content in the synovial membrane. The fibroblast-to-myofibroblast transition in arthrofibrosis is the outcome of former tissue surgery or injury [3, 6]. Myofibroblasts are the key cells in fibrogenesis which synthesize the extracellular matrix (ECM) components and provide structural support for the extracellular environment [7]. They possess a highly contractile phenotype marked by increased expression of myofibroblast markers alpha-smooth muscle actin ($\alpha$-SMA) and human xylosyltransferase-I (XT-I) [8–10].

Transforming growth factor-beta (TGF-$\beta$) is a key factor in fibrotic development in different organ systems and has been demonstrated to be highly expressed in arthrofibrotic tissues [11, 12]. It is involved in many profibrotic processes, including epithelial-to-mesenchymal transition and enhancing the expression of the tissue inhibitor of metalloproteinases (TIMP) or elevating ECM deposition [13]. The TGF-$\beta$ superfamily of cytokines includes bone morphogenetic proteins (BMP) and the TGF-$\beta$/activin subfamily, which activate distinct signaling pathways. Three TGF-$\beta$ isoforms: TGF-$\beta$1, TGF-$\beta$2 and TGF-$\beta$3 exist in mammals. They are encoded by three different genes: *TGFB1*, *TGFB2* and *TGFB3*, that are tissue-specifically expressed [14]. TGF-$\beta$1 is produced as a prohormone, consisting of a *C*-terminal mature TGF-$\beta$ and a *N*-terminal latency-associated protein. The activation of latent TGF-$\beta$ in the context of inflammation and fibrosis has been shown to involve integrin binding. Knocking out $\alpha$v$\beta$6/8 integrin subunits, such as $\beta$6, $\alpha$v or $\beta$8, encoded by the genes *ITGA5*, *ITGB6* and *ITGB8*, results in defects similar to the phenotype of TGF-$\beta$1 knockout mice [13]. In addition, proteases and glycoproteins, including plasmin, matrix metalloproteinase (MMP)-2 and thrombospondin (THBS)-1, are also involved in latent TGF-$\beta$ activation. Furthermore, the mechanical state of the ECM has been described to promote latent TGF-$\beta$ and fibroblast activation [13, 15, 16]. The canonical TGF-$\beta$ signaling pathway is mediated by SMAD proteins (SMAD2/3, SMAD4), which, upon ligand binding to cell surface receptors, transmit the signal to the nucleus. The target gene activation by an activated SMAD complex is dependent on DNA-binding cofactors, such as AP-1 and specificity protein-1 (SP-1), that provide a specific recognition of regulatory elements in target genes [17].

The action of TGF-$\beta$ depends on not only the cell type and its differentiation, but also the total cytokine milieu present. Therefore, dysregulation of this cytokine microenvironment might alter the TGF-$\beta$-mediated effects and contribute to disease development [14]. The cytokines TGF-$\beta$, IL-1$\beta$, and IL-6 are produced by fibroblasts and act on fibroblasts to promote inflammatory and fibrotic responses. Other important profibrotic soluble factors include connective tissue growth factor (CTGF) and members of the platelet-derived growth factor (PDGF) family, which have been shown to enhance the TGF-$\beta$-binding to its receptors or promoting cell proliferation and ECM synthesis [18–21].

The ECM that is synthesized in fibrotic remodeling serves as a scaffold for immune cells, fibroblasts and endothelial cells and provides a reservoir for growth factors, cytokines, and nutrients. It consists of different macromolecules that can be divided into collagenous and non-collagenous proteins [22]. The collagen family comprises 28 different types of collagens, encoded by more than 42 genes. Fibroblasts predominantly express type I, II, V and VI collagens [23]. Collagen neosynthesis correlates with the expression of the collagen cross-linking enzyme lysyl oxidase (LOX), which is induced by TGF-$\beta$1 [24]. The non-collagenous proteoglycans (PGs) make up a significant fraction of fibrotic lesion. The PGs are divided into four

groups based on their extracellular localization, size and structural properties: cell surface and membrane-bound PGs, such as syndecan (SDC-2), extracellular PGs, for example, versican (VCN) and aggrecan (ACAN), basement membrane PGs, including perlecan (HSPG-2), and small leucine-rich PGs, such as biglycan (BGN) and decorin (DCN) [25, 26]. Regarding structure, PGs are comprised of a single or multiple linear polysaccharides, the glycosaminoglycan (GAG) chains that are covalently attached to the PG core protein by a uniform tetrasaccharide linker.

The initial step in the posttranslational biosynthesis of the tetrasaccharide linker region is catalyzed by human XT-I and XT-II (EC 2.4.2.26), which are encoded by the genes *XYLT1* and *XYLT2*, respectively. The enzymes are localized in the Golgi apparatus and catalyze the transfer of D-xylose from donor substrate UDP-D-xylose to specific serine residues in the acceptor PG core protein. The enzymes were shown to be secreted into the extracellular space together with PGs, probably attached to the PG core protein [27, 28]. Although the biological role of XT secretion remains unknown, this feature makes it possible to determine the PG biosynthesis rate in organs and different cell models [28–30]. This points to the important role of XT-I as a disease modifier in pathologies characterized by an altered PG biosynthesis, such as systemic sclerosis (SSc) and arthrofibrosis [8, 31, 32]. Regarding arthrofibrosis, the *XYLT1* expression was used to control the efficiency of an anti-fibrotic treatment in a rat model of knee implant surgery or demonstrate the myofibroblast differentiation rate in human arthrofibrotic tissues [33, 34]. Since fibroblast-to-myofibroblast transition was determined by the increased expression and activity of XT-I [9, 35], the inhibition of XT could be a promising approach toward fibroproliferative diseases by modulating a downstream mediator of the TGF-β signaling [36].

Epigenetic alterations in myofibroblasts increase the activity of inflammatory and profibrotic genes in fibrosis and appear to serve as a type of memory of the insult [37]. Therefore, recent and ongoing clinical trials were initiated based on findings from *in vitro* studies using primary human fibroblasts [38]. This study employs two cell culture models to characterize the cellular and molecular phenotype of primary synovial fibroblasts from arthrofibrotic tissues (AFib). Cells were maintained under serum-reduced culture conditions in one experimental setup. This approach should identify potential differences in the myofibroblast phenotype of arthrofibrotic and non-arthrofibrotic synovial fibroblasts (SF), based on epigenetic changes that were maintained *in vitro* even in the absence of exogenous profibrotic mediators [37].

In addition, it was demonstrated previously that revision TKA accelerates the likelihood of developing arthrofibrosis [3]. We assume that this finding is based on the fact that myofibroblasts that have reverted to fibroblasts are more likely to become reactivated when exposed to further insults [39, 40]. We address this phenomenon with a second setup which mimics the revision TKA or repeated wound healing response by maintaining AFib and CF in serum-reduced growth medium supplemented with TGF-β1. Furthermore, using the downstream mediator XT-I of arthrofibrosis identified previously [8, 33, 34], this study aims to identify altered metabolic pathways and genes that may lead to the changes in XT-I expression and secretion in arthrofibrotic tissue remodeling observed in earlier studies [8, 34].

In summary, the overall aim of this study was to perform a cell biological characterization of Afib synovial fibroblasts and to provide a sound basis for more in-depth follow-up studies. As far as we know, this is the first published study dealing with the remodeling of the extracellular matrix of synovial Afib fibroblasts. The exploration and understanding of irregulated cellular pathways and mediators involved in the maintenance of the myofibroblast phenotype in AFib is crucial for developing new therapeutic interventions for arthrofibrosis and other fibroproliferative diseases in the future. The data generated by this study make a valuable contribution to deciphering the molecular mechanisms involved in arthrofibrosis.

## Materials and methods

### Primary cell culture models and sample preparation

Currently published experimental data in the field of arthrofibrosis research are mainly limited to complete human tissue samples or animal models. Cells in primary culture closely resemble the parental tissue from which they were isolated, but have a limited lifespan. Obtaining patient-derived primary cells from surgical material and testing them for suitability and function takes several months. Consequently, the usage of a limited number of primary cultures in this study is based on the small number of available and qualitatively suitable human primary cells in low passages from patients diagnosed with knee arthrofibrosis. Considering that sample size is limited, the main advantage of our primary cell culture data is that all future experimental arthrofibrosis studies can use and build on the established methodology and results presented here. This will not only save valuable material for the future, but will also lead to faster research progress.

Primary SF were provided by the authors of [8], who conducted the former preparation, characterization and conservation of the cells used in this study. All arthrofibrosis tissue donors were diagnosed based on histopathological findings and clinical parameters such as limited range of joint motion and the time between knee replacement therapy and diagnosis in the rehabilitation center. The clinical characteristics of the patients with arthrofibrosis are listed in our previous publication [8].

The SF were maintained under standardized conditions (5% $CO_2$, 37°C) in 100 x 20 mm cell culture dishes (Greiner bio-one, Frickenhausen, GER) with Dulbecco's modified eagle's medium (DMEM; Thermo Fisher Scientific, San Diego, USA) containing 10% (*v/v*) fetal calf serum (FCS; PAN biotech, Aidenbach, GER), 1% (*v/v*) Penicillin-Streptomycin-Amphotericin B solution (100x; PAN biotech, Aidenbach, GER) and 2% (*v/v*) L-glutamine (200 mmol/L; PAN biotech, Aidenbach, GER). A change of medium was performed twice a week. The SF from controls (CF) and arthrofibrosis patients (AFib) were subcultured by reaching cellular confluence of 80% using a solution of 0.05% (*w/v*) trypsin and 0.02% (*w/v*) EDTA (10x; PAN biotech, Aidenbach, GER) in Dulbecco's phosphate buffered saline (PBS, 1x; Thermo Fisher Scientific, San Diego, USA) and utilized at passages four to ten.

An earlier established fibrosis cell culture model was applied to investigate any putative differences between CF and AFib [8, 9]. Unless otherwise stated, SF were maintained on hard tissue culture substrates with a low density of 40 cells per $mm^2$ in fully supplemented DMEM for 24 h to promote their transdifferentiation to proto-myofibroblasts [8]. A serum withdrawal for 24 h was performed by decreasing the FCS quantity in the growth medium from 10 to 0.1% (*v/v*) to synchronize the cells and decrease FCS-mediated effects. Cells in the first experimental setup were maintained under those serum-reduced culture conditions for the time points indicated. In the second experimental setup, the profibrotic environment was mimicked by maintaining the cells in serum-reduced DMEM (0.1% (*v/v*) FCS) supplemented with 5 μg/L TGF-β1 (Miltenyi Biotech, Bergisch Gladbach, GER) for the time points indicated.

The cell culture supernatants were collected to analyze the extracellular XT activity at 24 or 96 h or to determine the presence of proinflammatory cytokines at 48 h. The cell lysates were prepared by incubating the cell monolayer with a fixed amount of lysis buffer provided by the RNA extraction kit (Macherey-Nagel, Düren, GER) for gene expression analysis at 2 or 48 h.

### Nucleic acid extraction and synthesis of complementary DNA

The extraction of RNA and DNA from cell lysates and the synthesis of complementary DNA (cDNA) using RNA as a template were performed as described previously [9, 35]. The quality

of the RNA extracted was determined by a 2100 Bioanalyzer (Agilent Technologies, Santa Clara, CA, USA) and quantified with the Nanodrop 2000 spectrophotometer (Thermo Fisher Scientific, San Diego, USA).

## Quantitative real time polymerase chain reaction analysis

The quantitative real time polymerase chain reaction (qRT-PCR) was performed according to our previous work using a SYBR green dye-based real time amplicon detection by a LightCycler480 Instrument II system (Roche, Basal, Switzerland) [35]. The gene-specific primer sequences used are listed in S1 Table. The gene expression levels were normalized to the geometrical mean of the expression levels of Glyceraldehyde 3-phosphate dehydrogenase (*GAPDH*), Hypoxanthine-guanine phosphoribosyltransferase (*HPRT1*) and β2 microglobulin (*B2M*), and quantified using the $\Delta\Delta C_T$ method considering the PCR efficiency of the genes unless otherwise stated [41]. The qRT-PCR analyses were performed in biological and technical triplicates for each donor-derived primary cell culture.

The $RT^2$ profiler human fibrosis PCR array (SABiosciences PAHS-120ZA, Qiagen, Hilden; GER) was used to examine the relative expression of 84 fibrosis-associated genes by qRT-PCR. Cell culture experiments were performed in biological triplicates for all donor-derived primary cell cultures and cytokine treatments. Following the RNA extraction using the NucleoSpin RNA kit (Macherey-Nagel, Düren, GER), cDNA synthesis was performed using the $RT^2$ first strand kit (Qiagen, Hilden, GER), according to the manufacturer's instructions. All three RNA samples of one donor or treatment were combined and utilized for the cDNA synthesis. The cDNA templates generated were each combined with a $RT^2$ SYBR Green qPCR mastermix (Qiagen, Hilden, GER) and loaded into the wells of the $RT^2$ Profiler PCR Array (4 x 96-well format). The PCR was performed according to the manufacturer's recommendation using a Roche LightCycler480 instrument. Relative gene expression was calculated by the $\Delta\Delta C_T$ method using the expression level of *B2M* for data normalization.

## Wound healing and migration assay

The wound healing and migration assay was performed as described previously [42]. In brief, $1\times10^6$ cells per 60 mm cell culture dish (Greiner bio-one, Frickenhausen, GER) were cultivated for 18 h to generate a confluent cell monolayer. A physical gap was created within the cell monolayer the next day and the process of cell migration into the gap was monitored over a period of 72 h using a JuLI Br live cell analyzer system (Peqlab, Erlangen, GER).

## Collagen gel contraction assay

The collagen gel contraction assay was carried out as previously described [42]. In short, $2.5\times10^5$ cells were cultured overnight on a preformed collagen gel (12-well format) that was prepared by neutralizing an acidic rat tail collagen type I solution (5 g/L; ibidi, Martiensried, GER) with a sodium hydroxide solution (Merck KGAA, Darmstadt; GER) in PBS (10x; Thermo Fisher Scientific, San Diego, USA). A serum withdrawal was performed the next day. After 24 h, cells were grown in serum-reduced DMEM supplemented with 5 µg/L TGF-β1 or vehicle. The collagen gels were released from the sides of the culture dishes to initiate contraction and the subsequent change in gel size was monitored at various time points. A maximum of 48 h after gel release was chosen for area quantification using ImageJ software (Version 1.48v, [43]) to avoid the influence of proliferative effects on the gel area measured. The gel contraction assay was performed with four biological replicates per donor-derived primary cell culture. Data are presented as the percentage of the initial gel size.

## Sirius red dye assay

The relative levels of insoluble collagen accumulation in SF cultures were determined by Sirius Red dye assay [44]. Sirius Red (Direct red 80; Sigma-Aldrich, St. Louis, Missouri, US) was dissolved at 0.1% (*w/v*) in saturated aqueous picric acid (Morphisto, Frankfurt a. M., GER). The fixative solution was prepared by combining 28% (*v/v*) ethanol (Merck, Darmstadt, GER), 4% (*v/v*) formaldehyde (Sigma-Aldrich, St. Louis, Missouri, US) and 2% (*v/v*) glacial acetic acid (Roth, Karlsruhe, GER) [45].

The SFs were cultured at a density of 180 cells per $mm^2$ (12-well format) in fully supplemented growth medium. A medium change was performed after 18 h, and cells maintained for an additional 144 h in complete growth medium supplemented with TGF-β1 (5 µg/L) or vehicle (PBS, 1x). Regarding Sirius Red staining, the cell monolayer was washed with PBS (1x) and incubated with fixative for 10 min. After repeating the washing of the cells with PBS (1x), the Sirius Red dye solution was added and incubated for 1 h. Thereafter, the staining solution was withdrawn, and all non-bound dye residues were removed by extensive washing of the cell layer with hydrochloric acid (0.01 M). The resulting cell morphology and staining was recorded by bright-field microscopy before resolving the dye. Quantification of the staining was performed by extracting the dye with sodium hydroxide (1 M) for 2 h. The eluted dye solution was collected and the OD values at 540 nm were determined in technical duplicate utilizing a Tecan reader infinite 200 PRO spectrophotometer (Tecan, Männedorf, Switzerland). The Sirius Red dye assay was performed in biological triplicate per donor-derived primary cell culture. Three additional biological culture replicates were made and used for DNA extraction for normalization purposes.

## Alcian blue dye assay

The PGs are polyanionic macromolecules that can be quantified by the usage of cationic dyes that bind to the highly negatively charged GAG structures via electrostatic interactions [46]. The PG content of primary cell cultures was determined using the cationic Alcian blue dye, and carried out analogously to previous studies [47]. Alcian blue G8X was purchased from Sigma-Aldrich (St. Louis, Missouri, US) to prepare a 1% (*w/v*) Alcian blue solution in 3% (*v/v*) glacial acetic acid, adjusted to pH 2.5 using glacial acetic acid. A fixative solution was prepared by combining 0.1% (*v/v*) glutaraldehyde (Sigma-Aldrich, St. Louis, Missouri, US) with PBS (1x).

Six biological replicates were prepared per primary cell culture to determine the relative PG or GAG content of AFib and control cultures. One triplicate was used for the staining procedure and the other triplicate for DNA extraction. The SFs were cultured at a density of 180 cells per $mm^2$ (12-well format) in fully supplemented growth medium. A medium change was performed after 18 h, and cells maintained for an additional 144 h in complete growth medium supplemented with TGF-β1 (5 µg/L) or vehicle (PBS, 1x). Regarding Alcian blue staining, the cell monolayer was washed with PBS (1x) and incubated with a fixative solution of glutaraldehyde (0.1% (*v/v*); Sigma-Aldrich, St. Louis, Missouri, US) in PBS (1x) for 20 min. Cells were washed three times with PBS (1x) and incubated with 3% (*v/v*) glacial acetic acid for 3 min. Thereafter, the cell monolayer was stained with the prepared Alcian blue solution for 18 h and, subsequently, washed to remove all unbound dye residues. The resulting cell morphology and staining was documented by bright-field microscopy before resolving the dye for quantification. Dye extraction was performed by incubating the stained cells with guanidine hydrochloride (6 M; Sigma-Aldrich, St. Louis, Missouri, US) under constant agitation at 150 rotations per minute for 24 h. The OD value at 595 nm of the dye solution obtained was determined in technical duplicates using a Tecan reader infinite 200 PRO spectrophotometer and normalized to the cellular DNA content.

## Radiochemical XT activity assay

Quantification of XT activity in cell culture supernatants of SF was performed by a radiochemical enzyme assay, as described previously [48], using three biological and three technical replicates per experiment. The method is based on the enzyme catalyzed incorporation of [$^{14}$C]-D-xylose from UDP-[$^{14}$C]-D-xylose (PerkinElmer, Foster City, CA, USA) into silk fibroin as an XT acceptor substrate [49]. The quantity of [$^{14}$C]-D-xylose incorporated in a defined period of 4 h was determined by liquid scintillation spectrometry using a Tri-Carb 2800 TR counter (PerkinElmer, Foster City, CA, USA). The XT activity is proportional to the quantified disintegrations per minute (dpm) of a sample and was referred to the sample DNA content for normalization.

## Statistical analysis

All data are presented as mean values with standard errors of the mean (SEM). Statistical analysis of the experimental conditions was evaluated by nonparametric two-tailed Mann-Whitney *U* test using GraphPad Prism 8.0 (GraphPad Software, La Jolla, CA, USA) software. The absence of Gaussian distribution was confirmed by computing the Shapiro-Wilk normality test. A probability *p* value of <0.05 was considered significant.

An RT$^2$ Profiler PCR array data analysis was performed using the SABiosciences PCR array data analysis template (Qiagen, Hilden; GER) and online via the PCR array data analysis webpage. Statistical significance required at least a twofold increase in relative gene expression levels.

It should be noted here that two control and two AFib monocultures were used for each of the experiments. Unfortunately, the availability of suitable and representative primary cell cultures from arthrofibrosis patients is severely limited by the low material availability and the time-consuming production of primary cultures. To increase the statistical power of the study results, it will be necessary to increase the number of primary cultures used in the future. Increasing the number of samples will lead to the results no longer being classified as preliminary, but as statistically sound. Nevertheless, this study reveals data that are of particular importance in guiding future experiments and supporting the associated economical use of valuable cellular material.

## Ethics statement

All tissue samples used for primary SF preparation were obtained from patients undergoing knee revision surgery. The collection of tissue samples was performed in accordance with the German Medical Devices Act (MPG, guideline 98/79/EG) for the collection of residual human material for the evaluation of the suitability of an in vitro diagnostic device (§24). Thus, informed consent was waived due to the use of surgical waste from routine laboratory diagnostics and knee replacement surgery.

# Results

## Myofibroblasts from arthrofibrosis tissue exhibit enhanced contractile abilities and a more pronounced temporal XT secretion rate compared to controls

Myofibroblasts are highly contractile matrix-producing cells characterized by increased α-SMA expression and XT-I secretion [9]. Since our previous study demonstrated that AFib cultured in the presence of the fibrotic inducer TGF-β1 possesses both a higher α-SMA protein expression and XT activity compared to CF, the purpose of this study was to establish whether

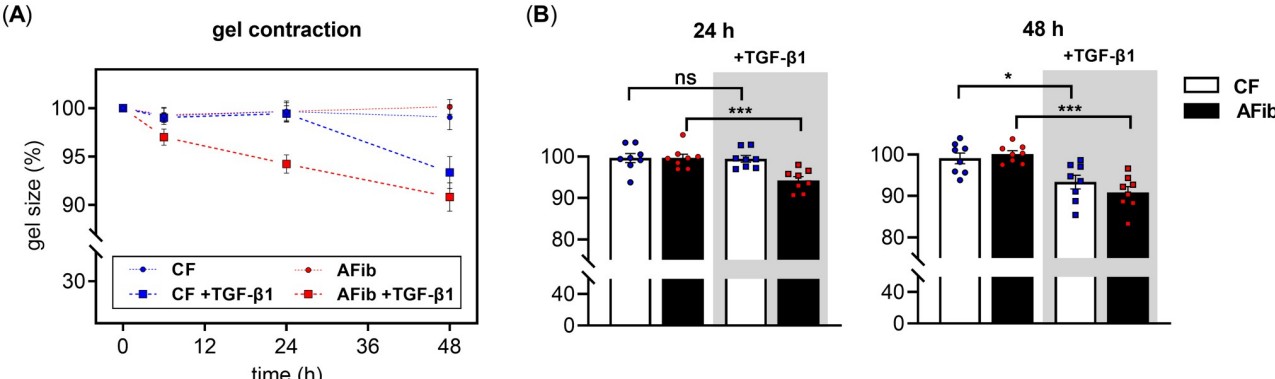

**Fig 1. Fibroblast-induced collagen gel contraction assay.** Human primary AFib (n = 2) and CF (n = 2) were cultured on preformed collagen gels the day before the experiment. Cells were serum-starved for 24 h and treated with TGF-β1 (5 µg/L) or vehicle (PBS, 1x). (A) Collagen gel contraction was monitored over a period of 48 h and presented as the percentage of the initial gel size. Data points shown are means ± SEM for four biological replicates per primary cell culture. (B) Bar chart display of the assay results measured 24 or 48 h post supplementation of AFib and CF with TGF-β1 (grey shaded) or vehicle. Columns represent means ± SEM for four biological replicates (data points) per primary cell culture. Mann-Whitney *U* test: not significant (ns), $p < 0.05$ (*), $p < 0.01$ (**), $p < 0.001$ (***).

this increase in myofibroblast content also impacted the wound healing properties and the temporal XT secretion rate of the cells.

We performed a collagen gel contraction assay to identify potential variations in the cellular contractility and fibroblast-matrix interaction of AFib and CF by culturing the cells on pre-formed collagen gel matrixes (Fig 1). The fibrotic microenvironment was mimicked in our *in vitro* model by TGF-β1-supplementation of the serum-reduced growth medium. The relative decrease in the surface area of the gel was monitored for 48 h and used as a parameter to quantify the degree of cell contractility (Fig 1A).

Quantification of the gel contraction was performed by comparing the initial with the final gel size after a fixed period during which the cells contract the collagen lattice. An incubation of 24 or 48 h was chosen to minimize the impact of cell expansion on the gel size quantified (Fig 1B). A significant decrease ($p < 0.001$) in gel size of 5.5 ± 1.3% (24 h) and 9.2 ± 1.6% (48 h) could be detected for the matrices containing AFib upon TGF-β1 supplementation. The CF containing gels with TGF-β1 did not show a significant gel size reduction after a cultivation period of 24 h compared to gels without TGF-β1 but did show a significant ($p < 0.05$) gel size reduction of 5.8 ± 2.1% after an incubation period of 48 h.

These findings suggest that TGF-β1 is a potent inducer of collagen gel contraction in AFib after a minimal incubation period of 24 h. In comparison to CF, AFib showed a temporally more pronounced TGF-β1-mediated cellular contractility.

We next analyzed the potential impact of autocrine cytokine loops that amplify and maintain the fibrogenic phenotype of AFib upon *in vitro* culture. Previous studies by our group demonstrated a time-dependent increase of extracellular XT activity that is further enhanced by the presence of profibrotic mediators [9, 35]. We analyzed the relative XT activity increase in the cell culture supernatant of AFib and CF between two time points using a radiochemical XT activity assay to obtain a rough indication of the potential cytokine actions in AFib (Fig 2).

We observed a higher time-dependent XT secretion rate in AFib (1.9 ± 0.3-fold, $p < 0.01$) compared to that of CF (1.4 ± 0.2-fold, $p < 0.05$) when grown under serum-reduced cell culture conditions (Fig 2A). Since the temporary XT activity increase observed in AFib may result from former *XYLT1* and *XYLT2* expression changes that were introduced *in vivo* and retained upon *in vitro* culture, we determined the basal *XYLT1* mRNA expression in AFib and CF

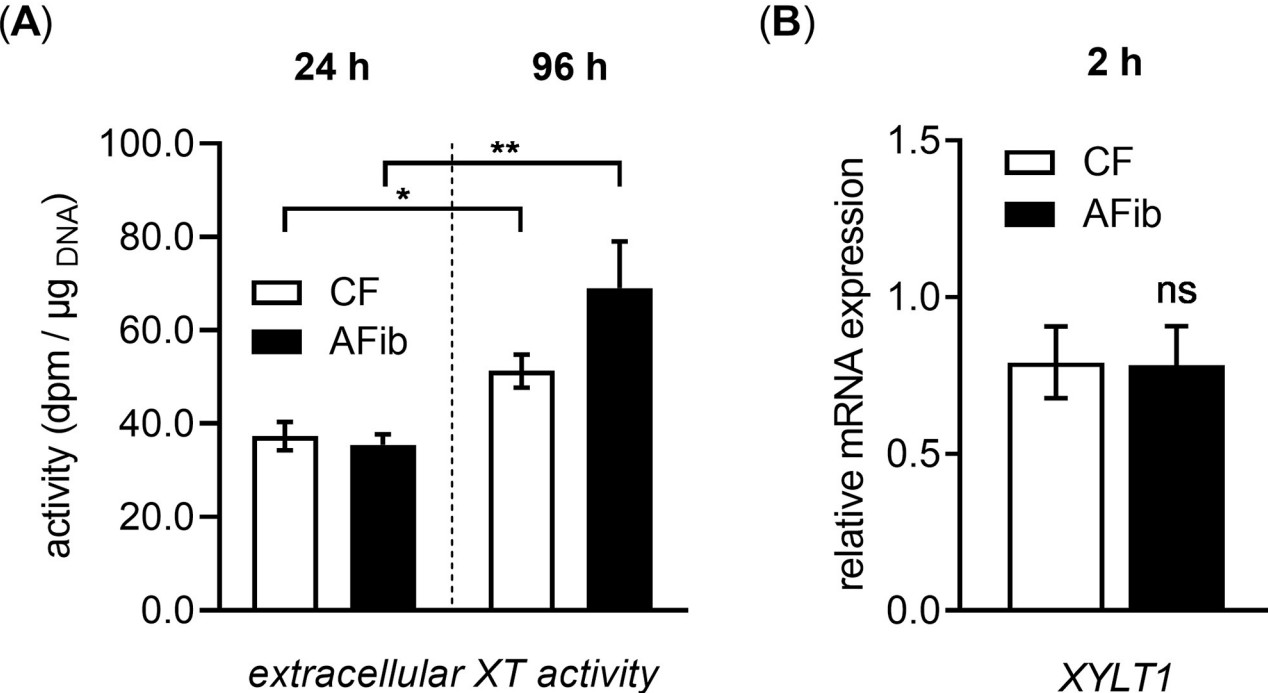

**Fig 2. AFib show a more pronounced temporal XT secretion rate than CF.** Human primary AFib (n = 2) and CF (n = 2) were cultured on the day before the experiment. Cells were serum-starved for 24 h and maintained under serum-reduced culture conditions for the time points indicated. (A) Cell culture supernatant was collected to determine the extracellular XT activity by radiochemical XT activity assay. Data shown are means ± SEM for three biological and two technical replicates per experiment. (B) Cell lysates were obtained and the relative *XYLT1* mRNA expression levels were determined by qRT-PCR. Expression levels were normalized to the expression level of *B2M*. Data shown are means ± SEM for three biological and three technical replicates per experiment. Mann-Whitney *U* test: not significant (ns), $p < 0.05$ (*), $p < 0.01$ (**).

upon a cultivation period of 2 h. This chosen culture period of 2 h was selected in analogy to previous work [9, 31, 35, 50], in order to exclude any *in vitro* cytokine-mediated effects on the expression changes determined. We found no significant *XYLT1* mRNA expression difference between AFib and CF (Fig 2B).

Therefore, we were able to exclude any former epigenetic changes affecting the extracellular XT activity of AFib. We conclude that the higher XT secretion rate observed in AFib relative to CF might be based on cytokine-mediated effects on the relative *XYLT1* expression.

### AFib show an aberrant *in vitro* collagen expression and deposition

In addition to the increased contractile properties and marker gene expression of overreactive myofibroblasts, these cells have been shown to generate and deposit excessive ECM proteins contributing to fibrotic conditions [15]. In order to verify this in our cell culture models and test whether AFib can retain their profibrotic phenotype upon *in vitro* culture, we analyzed the relative collagen deposition of AFib and CF by Sirius Red dye staining assay (Fig 3).

Without the presence of an additional TGF-β1 in the growth medium used, the relative collagen levels in AFib cultures were 1.2 ± 0.0-fold higher ($p < 0.0001$) than the levels in CF cultures after the same cultivation period of 144 h (Fig 3A). This relative difference was reverted by culturing AFib and CF in the presence of profibrotic mediator TGF-β1 for the period given (Fig 3B). This result shows that the increase in total collagen content in AFib could involve epigenetic alterations due to the former profibrotic microenvironment. Furthermore, it can be

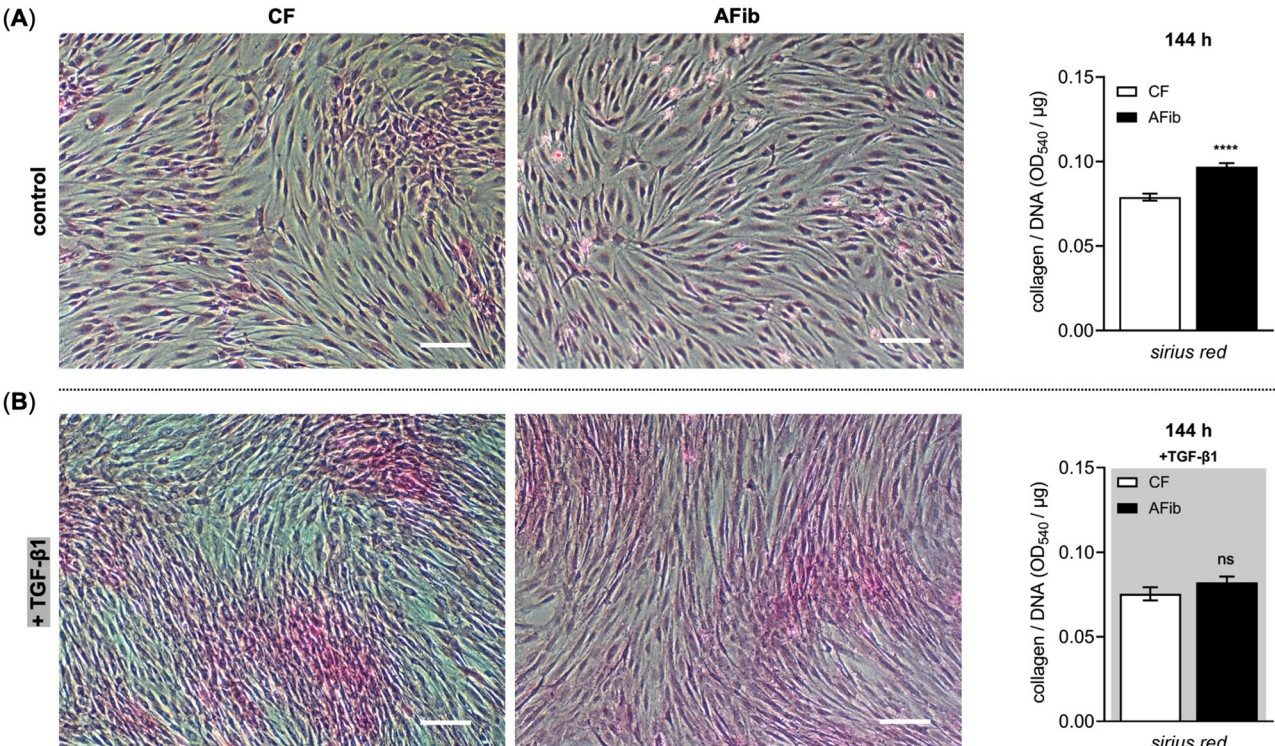

**Fig 3. Relative quantification of collagens deposited in AFib and CF by Sirius Red dye assay.** Human primary AFib (n = 2) and CF (n = 2) were cultured the day before the experiment. Growth medium was renewed, and cells maintained in fully supplemented growth medium for an additional 144 h (A) without or (B) with the presence of TGF-β1 (5 µg/L, gray shaded). Representative bright field images of the Sirius Red uptake of deposited collagen by Afib and CF monolayer cultures (scale bar: 100 µm). The *in situ* dye uptake by the cells following histological Sirius Red staining was determined via absorbance measurement at 540 nm ($OD_{540}$) after dye extraction. The OD values were referred to the samples' DNA content ($OD_{540}$/µg) for normalization. Data shown are means ± SEM for three biological replicates per donor-derived primary cell culture. Mann-Whitney *U* test: not significant (ns), $p < 0.0001$ (****).

concluded that the further TGF-β1 exposure of AFib *in vitro* did not alter their collagen accumulation relative to TGF-β1-treated CF.

In order to determine whether the increase observed in cellular collagen content in AFib was exerted at the transcriptional level, the mRNA expression of type I collagen (*COL1A1*) and type III collagen (*COL3A1*) in AFib and CF were analyzed by qRT-PCR. The qRT-PCR analysis was performed on reversed transcribed total RNA extracts of primary AFib and CF cultured in the presence or absence of TGF-β1 for a period of 48 h (Fig 4).

Using our serum-reduced cell culture model, we demonstrated a 2.2 ± 0.4-fold increased *COL1A1* mRNA expression ($p < 0.0001$) and a 1.5 ± 0.2-fold increased *COL3A1* mRNA expression ($p < 0.001$) in AFib compared to CF (Fig 4A and 4B). Consistent with the protein data, the collagen expression differences could be reversed by TGF-β1 supplementation (Fig 4D and 4E), indicating the contribution of autocrine cytokine-mediated effects on the former mRNA and protein expression increase observed in AFib. It can be concluded that the differences and similarities in the total collagen content of AFib and CF observed above are based on transcriptional changes of certain collagen species, such as *COL1A1* and *COL3A1*.

## Detection of GAG content and differentially expressed PGs genes in AFib

In addition to the fibrillar collagens, PGs are a class of non-fibrillar ECM proteins also known to accumulate in various fibrotic conditions [26]. In order to verify the initial assumption that

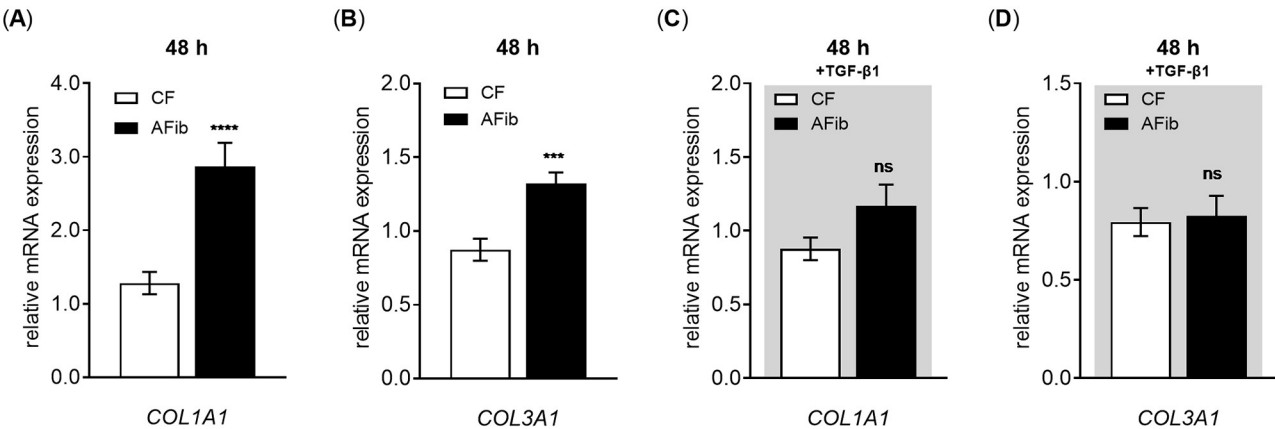

**Fig 4. Differences in basal *COL1A1* and *COL3A1* mRNA expression in AFib and CF.** Human primary AFib (n = 2) and CF (n = 2) were cultured the day before the experiment. Cells were serum-starved for 24 h and maintained in (A,B) serum-reduced or (C,D) TGF-β1 (5 μg/L, gray shaded) supplemented media for an additional 48 h. Relative expression levels of (A,C) *COL1A1* and (B,D) *COL3A1* were analyzed by qRT-PCR. The relative gene expression values were related to the respective CF treatments without (A,B) or with TGF-β1 (C,D). Data shown are means ± SEM for three biological and three technical replicates per experiment. Mann-Whitney *U* test: not significant (ns), $p < 0.05$ (*), $p < 0.001$ (***), $p < 0.0001$ (****).

AFib retain their profibrotic phenotype upon *in vitro* culture, we addressed the relative deposition of this non-fibrillar ECM protein class in our cell culture model by Alcian blue dye assay (Fig 5).

Without the presence of supplemented TGF-β1 in the growth medium used, the relative GAG level in AFib cultures was 1.3 ± 0.1-fold higher ($p < 0.0001$) than the level in CF cultures after the same cultivation period of 144 h (Fig 5A). This relative difference was still observable ($p < 0.01$) but less pronounced when culturing AFib and CF in the presence of the profibrotic mediator TGF-β1 for the period given (Fig 5B). Consistent with the collagen data reported previously, these results demonstrate an increase in the total GAG content in AFib that might be due to epigenetic alterations affecting the PG transcription level. In contrast to the collagen results observed, TGF-β1 exposure of AFib increased the total GAG content relative to TGF-β1-treated CF.

We evaluated the mRNA expression of a wide variety of PG core proteins by qRT-PCR to determine whether the overall GAG increase observed in AFib cultures was mediated by transcriptional changes of the PGs. We analyzed the PG gene expression of *ACAN*, *HSPG2*, *BGN*, *DCN* (Fig 6), *VCAN* and *SDC2* (S1 Fig).

Examination of primary AFib cultures that were maintained in cytokine-free growth medium for 48 h showed significant deviations in four of the six PG genes analyzed compared to primary CF cultures (Fig 6A–6D). We determined a relative increase in the mRNA expression of *ACAN* by 4.7 ± 0.7-fold ($p < 0.0001$), *HSPG2* by 1.9 ± 0.3-fold ($p < 0.001$) and *BGN* by 2.5 ± 0.3-fold ($p < 0.0001$) compared to the respective gene expression in CF (Fig 6A–6C). By contrast, the relative mRNA expression of *DCN* was 0.6 ± 0.1-fold decreased in AFib compared to CF (Fig 6D). The mRNA expression of *VCAN* and *SDC2* did not differ between AFib and CF that were cultured without TGF-β1 supplementation for 48 h (S1A and S1B Fig).

Interestingly, the deviations observed in PG expression in AF and CF was maintained in the cells cultured in the presence of TGF-β1 for 48 h (Fig 6E–6H). *ACAN* showed a relative increase by 3.4 ± 0.8-fold ($p < 0.0001$), *HSPG2* by 2.1 ± 0.3-fold ($p < 0.0001$) and *BGN* by 1.2 ± 0.1-fold ($p = 0.2$). Whereas the relative mRNA expression of *DCN* was 0.6 ± 0.1-fold ($p < 0.001$) decreased in AFib compared to CF (Fig 6H). The mRNA expression of *VCAN* and *SDC2* did not differ between AFib and CF that were cultured in the presence of TGF-β1 for 48

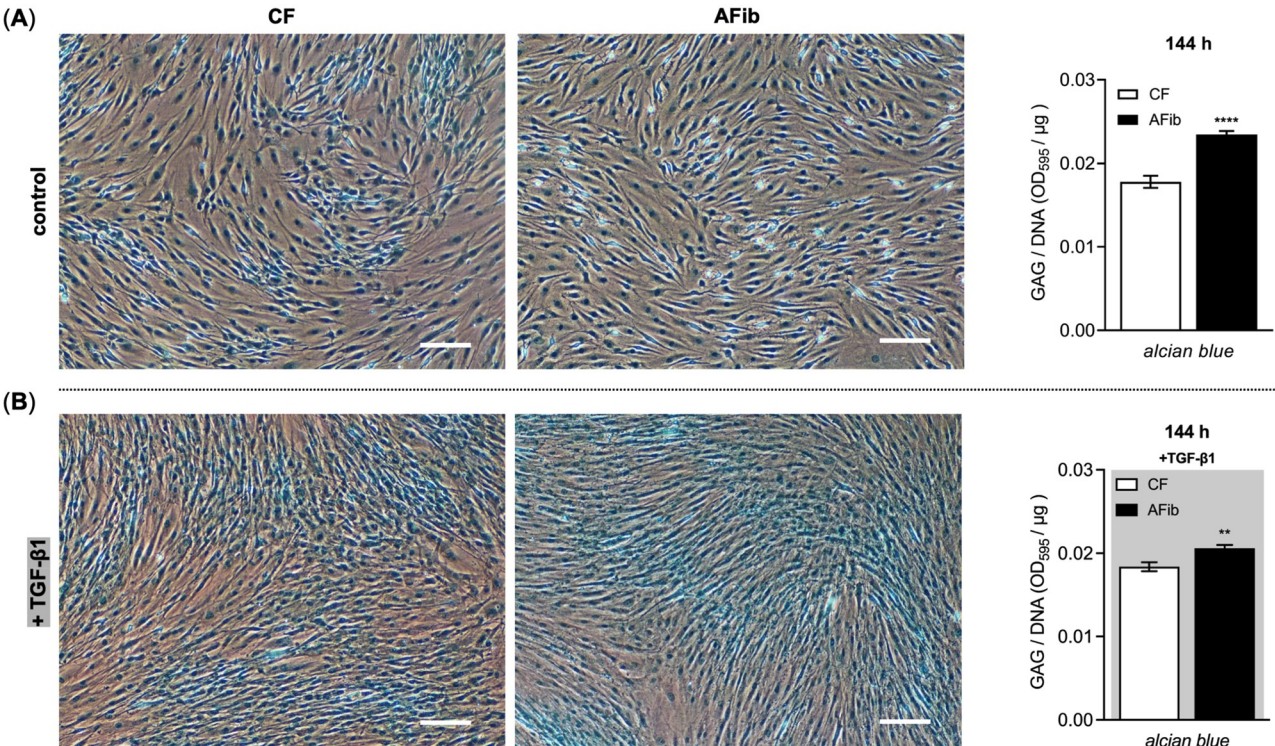

**Fig 5. Relative quantification of accumulated GAGs in AFib and CF cultures by Alcian blue dye assay.** Human primary AFib (n = 2) and CF (n = 2) were cultured the day before the experiment. Growth medium was renewed, and cells maintained in fully supplemented growth medium for an additional 144 h (A) without or (B) with the presence of TGF-β1 (5 μg/L, gray shaded). Representative bright field images of Alcian blue uptake of deposited GAGs by Afib and CF monolayer cultures (scale bar: 100 μm). The *in situ* dye uptake by the cells following histological Alcian blue staining is determined via absorbance measurement at 595 nm ($OD_{595}$) after dye extraction. The OD values were referred to the samples' DNA content ($OD_{595}$/μg) for normalization. Data shown are means ± SEM for three biological replicates per donor-derived primary cell culture. Mann-Whitney *U* test: $p < 0.01$ (**), $p < 0.0001$ (****).

h (S1C and S1D Fig). Since ACAN and HSPG2 are the most abundant PG in ECM-rich structures, such as cartilage or basement membrane [51], it can be concluded that the higher GAG content observed in AFib compared to CF might be based on the relative transcriptional increase of the PG encoding genes *ACAN* and *HSPG2*.

## AFib exhibit a fibrosis gene signature characterized by increased expression of pro-inflammatory mediators and TGF-β signaling components

Myofibroblasts are the key players in the pathogenesis of fibrosis in different organ systems [52], thus, exploration of different regulatory pathways involved in the maintenance of their phenotype is crucial for developing new therapeutic interventions. We performed a PCR array addressing the transcription level of 84 known fibrosis genes in AFib and CF to obtain an overview of the phenotype changes in AFib that were retained upon *in vitro* culture (Table 1).

Myofibroblasts obtained from tissue samples with arthrofibrosis display unique gene expression differences compared to those derived from control tissues (Table 1). Even in the absence of TGF-β1 supplementation, AFib displayed an increased expression of profibrotic genes *ACTA2* and *GREM1*, as well as ECM remodeling enzymes *LOX*, *MMP1* and *SERPINE1* compared to CF. Cell adhesion molecule *ITGB8* was also found increased in AFib compared to CF. Regarding inflammatory cytokines and growth factors, AFib showed an increased expression of *CCL11* and *IL1B*, along with *CTGF*, *EDN1* and *EGF* compared to CF. Regarding

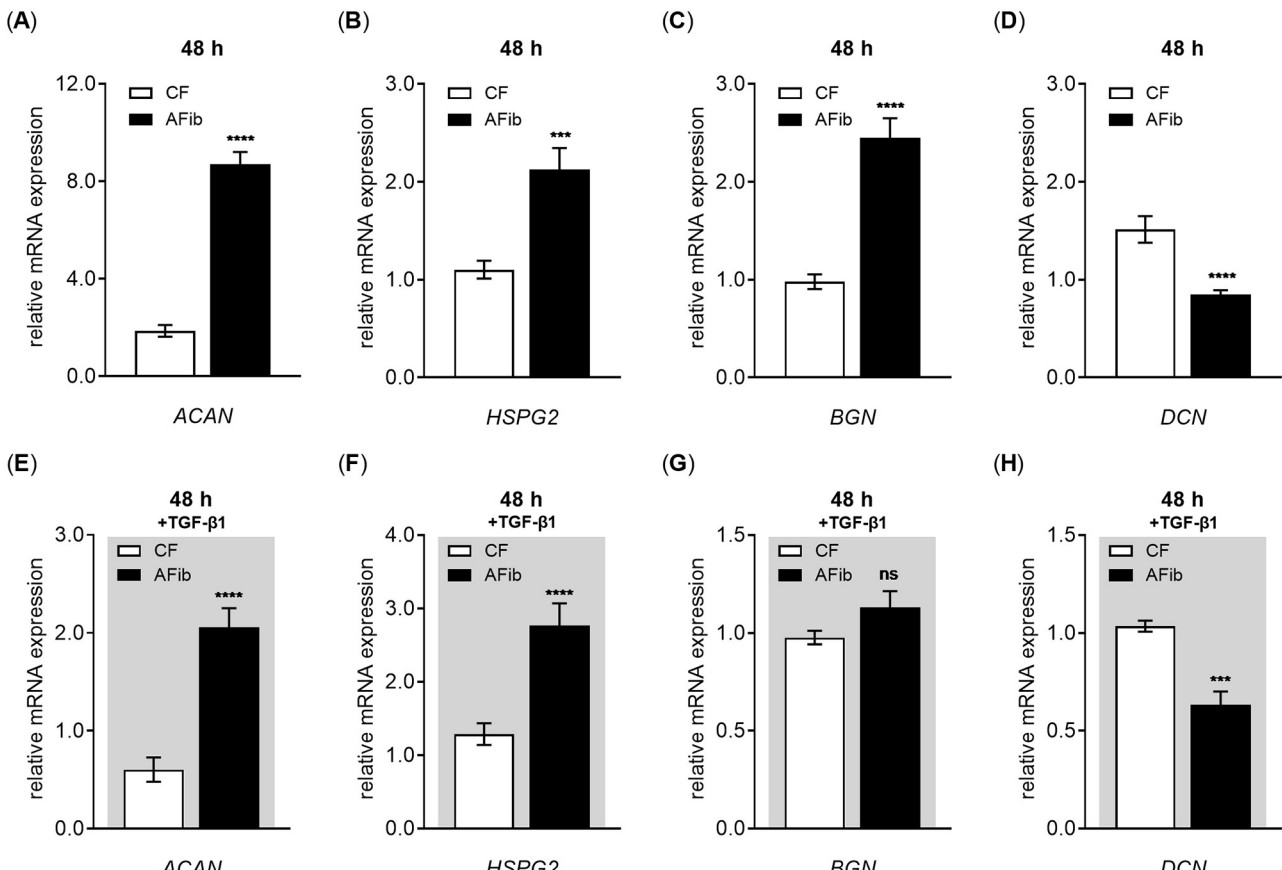

**Fig 6. Differences in basal and TGF-β1-induced PG mRNA expression in AFib and CF.** Human primary AFib (n = 2) and CF (n = 2) were cultured the day before the experiment. Cells were serum-starved for 24 h and maintained in (A–D) serum-reduced or (E–H) TGF-β1 (5 µg/L, gray shaded) supplemented media for an additional 48 h. Relative expression levels of the PG genes (A,E) *ACAN*, (B,F) *HSPG2*, (C,G) *BGN* and (D,H) *DCN* were analyzed by qRT-PCR. The relative gene expression values were related to the respective CF treatments without (A-D) or with TGF-β1 (E-H). Data shown are means ± SEM for three biological and three technical replicates per experiment. Mann-Whitney *U* test: not significant (ns), $p < 0.001$ (***), $p < 0.0001$ (****).

TGF-β superfamily members and the TGF-β signaling pathway, AFib showed a relative increase in *INHBE*, *SMAD3*, *TGFB2* and *THBS1* mRNA expression compared to CF. Furthermore, compared to CF, AFib showed a relative increase in *MYC* mRNA expression, another fibrosis-associated transcription factor. In addition to the relative gene expression increases found in AFib, we registered decreased transcriptional expressions of the fibrotic genes *MMP9*, *IL13*, *TNF*, *HGF* and *BCL2* in AFib relative to CF.

When culturing the cells in the presence of the profibrotic mediator TGF-β1, we found a relative increase in 6 of 13 increased genes in AFib that were not present in AFib cultured without TGF-β1. These genes included *AGT*, *MMP13*, *PLAU*, *ITGB6*, *IL1A* and *THBS2*. The other seven genes that were increased in AFib compared to CF comprise *ACTA2*, *GREM1*, *SERPINE1*, *EGF* and *TGFB2* and have been shown to overlap in both the TGF-β1-treated and -untreated group. We must consider that the *EGF* gene's average threshold cycle was relatively high corresponding to a relatively low expression level. Thus, the relative difference in *EGF* gene expression in AFib might be negligible.

In addition to the transcription increase in AFib detected in the presence of TGF-β1, we also found diminished gene expressions for *MMP3*, *MMP9* and *BMP7*, with the relative

**Table 1. Relative expression values of ECM and associated genes in AFib and CF generated by the RT$^2$ profiler human fibrosis PCR array.**

| Gene | − | + |
|---|---|---|
| **Profibrotic** | | |
| Smooth muscle actin alpha 2 (ACTA2) | 2.16 | 3.10 |
| Angiotensinogen (AGT) | 1.23[B] | 2.18[B] |
| Gremlin 1 (GREM1) | 3.26 | 6.21 |
| **ECM structural constituents** | | |
| Collagen type I alpha 2 (COL1A2) | 1.25 | 1.86 |
| Collagen type III alpha 1 (COL3A1) | 0.92 | 1.54 |
| Decorin (DCN) | 0.69 | 0.91 |
| **ECM remodeling enzymes** | | |
| Lysyl oxidase (LOX) | 2.07 | 1.91 |
| Matrix metallopeptidase 1 (MMP1) | 2.39[A] | 1.82[B] |
| Matrix metallopeptidase 13 (MMP13) | 0.63[B] | 2.20[B] |
| Matrix metallopeptidase 14 (MMP14) | 1.90 | 1.70 |
| Matrix metallopeptidase 2 (MMP2) | 0.79 | 1.14 |
| Matrix metallopeptidase 3 (MMP3) | 1.01 | 0.44 |
| Matrix metallopeptidase 8 (MMP8) | n.d.[C] | n.d.[C] |
| Matrix metallopeptidase 9 (MMP9) | 0.36[B] | 0.40[B] |
| Plasminogen activator, tissue type (PLAT) | 1.35[B] | 1.24[B] |
| Plasminogen activator, urokinase (PLAU) | 1.74 | 2.43 |
| Plasminogen (PLG) | n.d.[C] | n.d.[C] |
| Serpin family A member 1 (SERPINA1) | 1.02 | 1.26 |
| Serpin family E member 1 (SERPINE1) | 3.03 | 2.59 |
| Serpin family H member 1 (SERPINH1) | 1.39 | 1.28 |
| TIMP metallopeptidase inhibitor 1 (TIMP1) | 0.99 | 0.82 |
| TIMP metallopeptidase inhibitor 2 (TIMP2) | 1.38 | 1.80 |
| TIMP metallopeptidase inhibitor 3 (TIMP3) | 1.27 | 1.99 |
| TIMP metallopeptidase inhibitor 4 (TIMP4) | 1.11[B] | 1.17[B] |
| **Cell adhesion molecules** | | |
| Integrin subunit alpha 1 (ITGA1) | 1.04 | 1.45 |
| Integrin subunit alpha 2 (ITGA2) | 0.60 | 0.63[A] |
| Integrin subunit alpha 3 (ITGA3) | 1.37 | 1.96 |
| Integrin subunit alpha V (ITGAV) | 1.33 | 1.67 |
| Integrin subunit beta 1 (ITGB1) | 1.33 | 1.70 |
| Integrin subunit beta 3 (ITGB3) | 1.08 | 1.72 |
| Integrin subunit beta 5 (ITGB5) | 0.87 | 1.36 |
| Integrin subunit beta 6 (ITGB6) | n.d.[C] | 2.70[B] |
| Integrin subunit beta 8 (ITGB8) | 3.00 | 2.36[B] |
| **Inflammatory cytokines and chemokines** | | |
| C-C motif chemokine ligand 11 (CCL11) | 9.71[A] | 2.98[A] |
| C-C motif chemokine ligand 2 (CCL2) | 1.09 | 0.50 |
| C-C motif chemokine ligand 3 (CCL3) | 1.56[B] | n.d.[C] |
| C-C motif chemokine receptor 2 (CCR2) | n.d.[C] | n.d.[C] |
| C-X-C motif chemokine receptor 4 (CXCR4) | n.d.[C] | n.d.[C] |
| Interferon gamma (IFNG) | n.d.[C] | n.d.[C] |
| Interleukin 10 (IL10) | n.d.[C] | n.d.[C] |
| Interleukin 13 (IL13) | 0.45[B] | n.d.[C] |

*(Continued)*

**Table 1.** (Continued)

| Gene | − | + |
|---|---|---|
| *Interleukin 13 receptor subunit alpha 2 (IL13RA2)* | 0.73 | 1.14 |
| *Interleukin 1 alpha (IL1A)* | 1.36[B] | 3.21[B] |
| *Interleukin 1 beta (IL1B)* | 2.14[B] | 1.60[B] |
| *Interleukin 4 (IL4)* | 0.54[B] | 1.87[B] |
| *Interleukin 5 (IL5)* | 1.68[B] | 1.09[B] |
| *Integrin linked kinase (ILK)* | 1.1 | 1.9 |
| *Tumor necrosis factor (TNF)* | 0.46[B] | n.d.[C] |
| **Growth factors** | | |
| *Connective tissue growth factor (CTGF)* | 2.29 | 1.47 |
| *Endothelin 1 (EDN1)* | 2.18 | 1.72 |
| *Epidermal growth factor (EGF)* | 3.14[B] | 5.53[B] |
| *Hepatocyte growth factor (HGF)* | 0.30[B] | n.d.[C] |
| *Platelet derived growth factor subunit A (PDGFA)* | 1.84[A] | 1.86[A] |
| *Platelet derived growth factor subunit B (PDGFB)* | n.d.[C] | n.d.[C] |
| *Vascular endothelial growth factor A (VEGFA)* | 1.07 | 1.96 |
| **TGF-β superfamily members and signaling pathway** | | |
| *Bone morphogenetic protein 7 (BMP7)* | n.d.[C] | 0.01[A] |
| *Caveolin 1 (CAV1)* | 1.11 | 1.73 |
| *Endoglin (ENG)* | 1.47 | 1.30 |
| *Inhibin beta E subunit (INHBE)* | 5.69[B] | 1.60 |
| *Latent transforming growth factor beta binding protein 1 (LTBP1)* | 0.69 | 0.94 |
| *SMAD family member 2 (SMAD2)* | 1.11 | 1.75 |
| *SMAD family member 3 (SMAD3)* | 2.07[B] | 1.07[B] |
| *SMAD family member 4 (SMAD4)* | 1.15 | 1.54 |
| *SMAD family member 6 (SMAD6)* | 1.21[A] | 0.96[B] |
| *SMAD family member 7 (SMAD7)* | 1.78 | 1.54 |
| *Transforming growth factor beta 1 (TGFB1)* | 1.75 | 1.37 |
| *Transforming growth factor beta 2 (TGFB2)* | 2.00 | 3.56 |
| *Transforming growth factor beta 3 (TGFB3)* | 1.04[A] | 0.71[A] |
| *Transforming growth factor beta receptor 1 (TGFBR1)* | 1.14 | 1.62 |
| *Transforming growth factor beta receptor 2 (TGFBR2)* | 0.67 | 1.59 |
| *TGFB induced factor homeobox 1 (TGIF1)* | 1.31 | 1.53 |
| *Thrombospondin 1 (THBS1)* | 3.88 | 1.63 |
| *Thrombospondin 2 (THBS2)* | 0.90 | 2.69 |
| **Other fibrosis-associated transcription factors** | | |
| *CCAAT enhancer binding protein beta (CEBPB)* | 1.18 | 1.32 |
| *Jun proto-oncogene, AP-1 transcription factor subunit (JUN)* | 1.71 | 1.67 |
| *MYC proto-oncogene, bHLH transcription factor (MYC)* | 2.15 | 1.59 |
| *Nuclear factor kappa B subunit 1 (NFKB1)* | 1.56 | 1.55 |
| *Specificity protein 1 (SP1)* | 1.07 | 1.49 |
| *Signal transducer and activator of transcription 1 (STAT1)* | 1.32 | 1.11 |
| *Signal transducer and activator of transcription 6 (STAT6)* | 0.77 | 1.04 |
| **Epithelial-to-mesenchymal transition (EMT)** | | |
| *AKT serine/threonine kinase 1 (AKT1)* | 1.50 | 1.52 |
| *Snail family transcriptional repressor 1 (SNAIL1)* | 0.88[B] | 0.93[B] |
| *BCL2, apoptosis regulator (BCL2)* | 0.20[B] | 1.52[B] |

(*Continued*)

**Table 1.** (Continued)

| Gene | − | + |
|------|---|---|
| *Fas ligand (FASL)* | n.d.[C] | n.d.[C] |

The AFib (n = 2) and CF (n = 2) were cultivated (40 cells/mm²) in the presence (+) and absence (−) of TGF-β1 (5 μg/L) for 48 h. The fold change (FC) values describe the difference between CF and AFib that were cultured without (−) or with (+) TGF-β1. FC values greater than one indicate an up-regulation, whereas FC values less than one indicate a down-regulation. The FC values greater than 2 are indicated in red; FC values less than 0.5 are indicated in blue.

[A.] This gene's average threshold cycle is relatively high (>30) in either the control or the test sample and is reasonably low in the other sample (<30). These data mean that the gene's expression is relatively low in one sample and reasonably detected in the other, suggesting that the actual fold-change value is at least as large as the calculated and reported fold-change result. This fold-change result may also have greater variations if the p value >0.05; therefore, it is important to have enough biological replicates to validate the result for this gene.

[B.] This gene's average threshold cycle is relatively high (>30), meaning that its relative expression level is low in both control and test samples, and the p-value for the fold-change is either unavailable or relatively high (p > 0.05). This fold-change result may also have greater variations; therefore, it is important to have enough biological replicates to validate the result for this gene.

[C.] This gene's average threshold cycle is either not determined or greater than the cutoff defined (≥35) in both samples, meaning that its expression was undetected, making this fold-change result erroneous and uninterpretable.

decrease in *MMP9* mRNA expression overlapping in both the TGF-β1-treated and -untreated group. These results indicate novel differences in gene expression of myofibroblasts obtained from arthrofibrosis tissues that could be further validated by protein analysis or used for comparable purposes with other fibrosis conditions.

As is known from previous studies, IL-6 is capable of inducing cell migration during tissue remodeling in a wide variety of human cell types [53–55]. In order to explore the potential contribution of secreted cytokines or chemokines on the cell migration of AFib and CF, we conducted a wound healing and migration assay (Fig 7).

Wound healing was mimicked by inserting a defined scratch into the confluent cell monolayer. Graphical evaluation revealed no differences in the random cell migratory capacity of AFib and CF, both reaching total wound closure after 60 h of cell maintenance. Since the migration speed of cells is determined by the presence of certain cytokines, we conclude from this initial data that the autosecreted pro-inflammatory cytokines present in the cell culture supernatant of AFib did not affect the relative migratory capacity of the cells.

## Discussion

Arthrofibrosis is a fibroproliferative disorder that is characterized by the excessive production of ECM proteins in a joint [5]. Although there are few studies on the pathogenesis and molecular mechanism of arthrofibrosis compared to other fibrotic diseases [56], there might be common or unique pathogenic pathways that can be used for developing targeted treatment approaches. Among the disease effector cells, myofibroblasts are thought to play a key role in the development of arthrofibrosis [10]. While primary human cells from patients are considered as a unique resource to understand human disease biology more comprehensively [38], recent arthrofibrosis research has mainly been limited to animal models or the usage of complete human tissues that do not differentiate between specific cell types [33, 56, 57]. Therefore, the goal of our study was to provide further insights into the arthrofibrotic phenotype of human SF obtained from arthrofibrosis lesions, providing a basis for the future development of fibroblast-targeted therapeutic interventions. Given the limited availability of primary cell

## cell migration

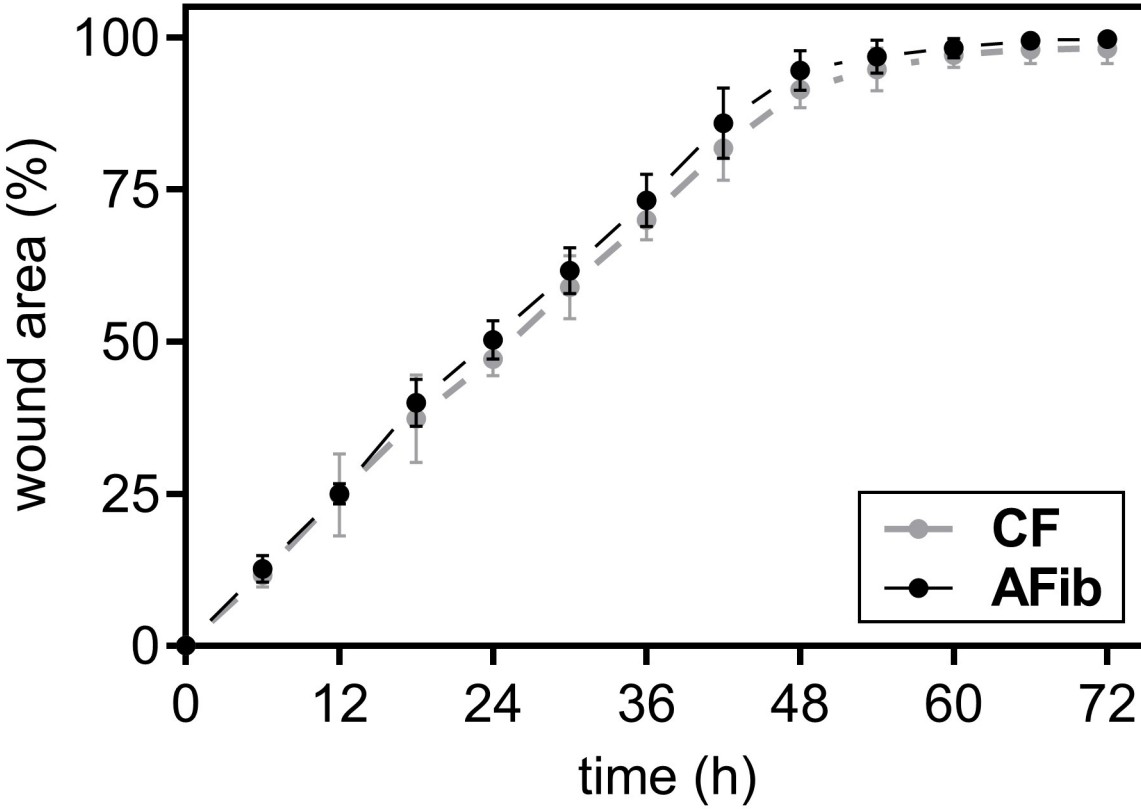

**Fig 7. Cell migration of AFib and CF did not differ from each other.** Human primary AFib (n = 2) and CF (n = 2) were cultured the day before the experiment. A physical gap was created by the mechanical scratching of the confluent monolayer, and the process of cell migration into the gap was observed over a period of 72 h using a live cell analyzer system and graphical analysis software. Data are means ± SEM of two biological replicates per donor-derived primary cell.

cultures from arthrofibrosis patients, publication of experimental studies may lead to the scientific community benefiting from the results presented, further accelerating scientific progress.

Fibroblasts isolated from fibrotic tissues are characterized by increased XT-I secretion and higher α-SMA levels, leading to an increased contractility [9, 58]. By using a myofibroblast cell culture model with AFib, Faust and colleagues [8] have shown that the basal α-SMA protein expression in AFib quantified by immunohistochemistry after 120 h was significantly increased compared to CF. Furthermore, the amplitude of XT activity increase in response to TGF-β1 was higher in AFib compared to CF, although no differences in the basal extracellular activity could be observed [8]. In continuation of the previous study [8], the present study aims to further analyze the myofibroblast phenotype of AFib by investigating the contractile abilities and temporal XT secretion rate of the cells. We found that in addition to the higher basal expression of *ACTA2* in AFib compared to CF determined by the PCR array, AFib possess a temporally more pronounced cellular contractility and a higher XT secretion rate compared to CF. The results are in line with the data from human skin fibroblasts showing a correlation between α-SMA expression and extracellular XT activity increase [9, 35]. It can be

concluded that the relative α-SMA expression can be utilized as an indicator for the relative myofibroblast content, whereas the temporal XT activity increase resembles a marker for the fibroblast-myofibroblast transition rate in AFib. This consumption is further strengthened by findings from arthrofibrotic biopsies of knee joints, showing a significant positive correlation of XT-I expression with the increasing number of fibroblasts, especially in the synovialis region [34].

The TGF-β1 and other members of the TGF-β1 superfamily have been shown to be potent inducers of *XYLT1* mRNA expression and XT activity in human dermal and SFs *in vitro* [8, 35]. Based on prior publications [31, 32, 50], it is likely that the *in vivo* exposure of these cytokines and growth factors could have led to epigenetic changes, including miRNA expression and histone modifications, affecting the relative basal expression of *XYLT1* in AFib and CF. It was shown that an increase in extracellular XT activity correlates with a former *XYLT1* mRNA increase [9, 30, 35]. Therefore, the changes of the basal *XYLT1* mRNA expression of AFib and CF could have contributed to the higher XT secretion rate in AFib observed in this study. This hypothesis was addressed by maintaining the cells in serum-reduced growth medium for a significantly shorter incubation time to exclude autocrine effects. Using this experimental setup, the impact of a higher basal *XYLT1* mRNA expression on the increased XT secretion rate in AFib observed in this study could be neglected. Another effect that may contribute to the fibrotic phenotype of AFib are positive feedback loops mediated by autocrine factors [12, 59]. The study by Bayram and colleagues [56] compared tissue samples from patients undergoing RTKA-A and primary TKA, which resembles our basal expression comparison of AFib and CF without TGF-β1 supplementation, and recognized a slight increase in *TGFB1* expression (1.75-fold) in RTKA-A. The results of our fibrosis PCR array also showed a slight increase in the expression of *TGFB1* (1.7-fold) in untreated cells. Regarding myofibroblast marker human XT-I [8, 9], known cytokines that are involved in the regulation of *XYLT1* mRNA expression or XT activity in primary cells comprise TGF-β1, activin A, IL-1β and thrombin. In addition, CTGF and EDN-1 were shown to modulate the TGF-β1-mediated *XYLT1* mRNA expression or XT activity in human dermal fibroblasts [30, 31, 35], whereas TGF-β2 has been shown to induce the *xylt1* gene transcription in primary astrocytes [60]. It can be concluded from our PCR array data that a significant proportion of known XT-I inducible or modifying factors, such as *TGFB1*, *IL1B*, *CTGF*, *EDN1* and *TGFB2*, were transcriptionally increased in AFib compared to CF. These results provide support for the hypothesis that cytokine-mediated feedback loops might be involved in the maintenance of the fibrotic phenotype of AFib upon *in vitro* culture. Consistent with our results, an increased expression of *IL1B*, *CTGF* and *EDN1* has also been found in arthrofibrotic tissue samples of human and animal origin [6, 56, 61]. By contrast, the expression differences in the TGF-β isoform *TGFB2* have not been reported in the context of arthrofibrosis.

It has been shown on the cellular level that TGF-β1 stimulates the transcriptional expression of different collagen species in SF obtained from either arthrofibrotic or healthy tissue [2, 8]. Using our primary human AFib and CF cell culture model, we observed the same results, showing that TGF-β1 is a potent inducer of *COL1A1* and *COL3A1* expression. However, conflicting results were found regarding the basal collagen gene transcripts in primary AFib and CF from our previous work and data from the literature using human and animal tissue samples [8, 57, 61]. In brief, our previous work did not observe a significant *COL1A1* and *COL3A1* mRNA expression difference in AFib and CF after a cultivation of 48 h under serum-reduced culture conditions [8]. This contrasts with the data from arthrofibrosis tissue of human origin by Bayram and colleagues, who found increased *COL1A1* and *COL3A1* expression in arthrofibrosis tissues compared to non-arthrofibrosis tissues determined by RNA sequencing [56]. In order to shed light on these contradictory findings, we analyzed the basal *COL1A1* and

*COL3A1* mRNA expression by qRT-PCR in our primary human AFib and CF cell culture model. Similar to the findings in arthrofibrotic tissue biopsies by Bayram and colleagues [56], our cultured primary AFib showed a basal *COL1A1* and *COL3A1* expression increase compared to CF. Interestingly, the *COL1A1* and *COL3A1* expression difference in our cell culture model was diminished in the presence of TGF-β1. We performed a Sirius Red dye assay to validate our results on the protein level and found a relative increase in total collagen content in AFib compared to CF that was also reversed in the presence of TGF-β1. These results are similar to previous immunohistochemical investigations, which showed increased tissue collagen concentration in patients undergoing revision TKA with non-arthrofibrotic origin (RTKA-NA) and RTKA-A compared to primary TKA [3]. No significant differences in the total collagen concentration and the collagen type I and type III content between patients undergoing non-arthritic revision TKA and patients undergoing arthrofibrotic revision TKA were found [3]. The usage of TGF-β1 in the cell cultivation of AFib and CF here, resembles the comparison mentioned above between patients undergoing non-arthritic revision TKA and patients undergoing arthrofibrotic revision TKA. In agreement with the results mentioned above [3], we found no difference in the total collagen content by Sirius Red assay between AFib and CF in the presence of TGF-β1 that could be due to the lack of *COL1A1* and *COL3A1* expression differences in the cells. It can be concluded that the single usage of collagen expression and protein content is not enough to characterize arthrofibrotic changes due to its high sample variability. Consistent with the sample variability in collagen expression analysis mentioned previously, the fibrosis PCR array performed did not depict pronounced changes (> 2-fold) on the *COL1A2* and *COL3A1* mRNA expression in AFib and CF. Our results are further strengthened by the fact that the rate of tissue remodeling in the late state of arthrofibrosis remains less pronounced than in the active state [62]. Therefore, it can be concluded that surgery-initiated collagen accumulation contributes to the myofibroblast phenotype during tissue remodeling regardless of clinical diagnosis.

Earlier studies have reported the simultaneously secretion of XT into the extracellular space with chondroitin sulfate PG during fibrotic tissue remodeling processes [27, 28]. To date, no study has evaluated the relation between the increased XT secretion rate and GAG content in AFib. Our gene profiling studies showed a significant upregulated basal expression of the extracellular PG genes *ACAN*, *HSPG2* and *BGN* and a significant decrease of relative *DCN* expression in AFib compared to CF. The mRNA expression of the PG genes *ACAN*, *HSPG2* and *DCN* in the presence of TGF-β1 remain significantly altered between AFib and CF. No expression differences were detected for *VCAN* and *SDC2* in AFib and CF. These results were validated by histological staining that shows a significant overall increase of accumulated GAGs in AFib cultures compared to CF regardless of whether TGF-β1 was supplemented to the growth medium or not. It can be concluded that the increased XT secretion rate in AFib is closely related to a higher PG expression and GAG content in AFib. The *ACAN* and *BGN* expression increase in AFib observed is consistent with data from patients undergoing RTKA-A [56]. The rest of our findings differ from those obtained from the prototypic fibrosis disease SSc, demonstrating an increase in *DCN* and *VCN* mRNA expression and a decrease in *BGN* mRNA expression in SSc fibroblasts [63]. Conversely, the *DCN* expression decrease in AFib observed agrees with the study by Abdel and colleagues [64], who showed a *DCN* mRNA expression decrease using a rabbit joint fibrosis model. It is noteworthy that in addition to differences in culture media conditions and cell culture models, fibroblasts possess topographic differences in the expression of genes related to growth and differentiation, ECM production and cell migration depending on their former anatomic sites [65–67]. The increase of a certain gene product could, therefore, have different consequences for the organism and disease outcome [68]. Regarding arthrofibrosis, studies using human fibroblasts have shown that cell

treatment with decorin reduces the proliferation of AFib and downregulates the expression of fibrotic markers [69, 70]. Thus, our data provide support for the usage of decorin in fibroproliferative diseases that are characterized by a decrease of *DCN* expression in the key effector cells.

Similar to findings from arthrofibrotic tissue biopsies [56, 61], cultured primary AFib also show a profibrotic phenotype at the transcriptional level of profibrotic genes and those involved in ECM remodeling, cell adhesion, inflammatory cytokines and chemokines, growth factors and TGF-β signaling, irrespective of additional TGF-β1 mediated effects. In addition to the increased expression of *ACTA2* that resembles previous findings [8, 57], we detected a relative increase in the profibrotic *GREM1* expression in AFib compared to CF. The expression of *GREM1* in several murine fibrosis models is greatly increased, while BMP signaling is decreased. Furthermore, administration of BMP-7 can decrease fibrosis in some models [71–73]. Consistent with this, the PCR array also depicted a relative decrease in *BMP7* expression in our arthrofibrosis cell culture model.

Basal mRNA expression increases of genes encoding ECM remodeling enzymes, such as *LOX*, *MMP1* and *SERPINE1*, were greatly increased in AFib compared to CF that were cultured in the presence or absence of TGF-β1. These findings are comparable to those of Bayram and colleagues [56], who suggested the extracellular *LOX* in arthrofibrosis remodeling. Furthermore, they identified *SERPINE1* expression to be higher in tissue biopsies of patients with arthrofibrosis compared to non-arthrofibrotic biopsies. The higher *MMP1* expression in AFib represented an intriguing finding, since MMP1 represents a collagen-degrading enzyme. This contradiction was also found in the transcriptional and immunohistochemical analysis of fibrotic lungs, revealing significantly high MMP1 expressions [68]. Nevertheless, the balance between ECM synthesis and degradation depends on the interaction between MMPs and TIMPs and their relative activities in the extracellular space [74]. As we saw a relative increase in total collagen content in AFib compared to CF, we suggest the results of the *MMP* expressions to be further validated on the protein or activity level of the enzymes.

Regarding the class of cell adhesion molecules that were shown previously to be increased in human arthrofibrotic tissues [56], we found increased transcriptional activity of *ITGB8* in primary AFib cultures. The αvβ8 integrin is a receptor for ECM-bound forms of latent TGFβ proteins and, therefore, promotes the activation of TGF-β signaling pathways [75]. As far as we know, aberrant *ITGB8* mRNA expression has not yet been described in the context of arthrofibrosis. Since it was shown previously that the IL-1β treatment of murine lung fibroblasts induces a sustained induction of collagen expression that is dependent on *Itgb8* and TGF-β [76], we assume that further analysis of this aberrant *ITGB8* mRNA expression in AFib would increase the mechanistic understanding of the molecular changes during arthrofibrosis.

In addition to the inflammatory cytokines and profibrotic growth factors mentioned above, we found a significant induction of glycoprotein *THBS1* and chemokine *CCL11* mRNA expression in AFib compared to CF in both culture conditions. An increase in the *THBS1* expression was also found in the study between samples of RTKA-A and RTKA-NA mentioned above and in an arthrofibrosis rat model [6]. To the best of our knowledge, this cytokine has not been reviewed in the context of arthrofibrosis before. A previous study showed that CCL11 has a direct and selective profibrogenic effect on human lung fibroblasts, enhancing their collagen synthesis and migratory ability, although it did not influence the expression of the profibrotic gene *ACTA2* or the cell-mediated collagen gel contraction [77]. Since we did not observe an increased migratory ability in our cell culture model, we presume that the increased expression of CCL11 affects the immune cell response rather than the myofibroblast characteristics of AFib itself. Nonetheless, this hypothesis should be further investigated in future experiments.

In summary, our study reveals a number of hitherto unreported information about the remodeling of the extracellular matrix in AFib. In particular, the establishment of an efficient technology for growing primary cultures and the identification of potential therapeutic targets make a valuable contribution to the deciphering of the biological processes that are abnormal in arthrofibrosis.

In spite of the fact that the present study has several limitations to consider, we have already pointed out these limitations in the section on materials and methods in order to be aware of them and to classify them properly. As the main findings of our study were from cell culture experiments with a limited number of donor-derived tissue samples, a larger confirmatory study is needed to verify the results of this initial hypothesis-generating study. In addition, we found that the amplitude of extracellular XT activity increase in response to TGF-β1 was higher in AFib compared to CF. However, we did not address the effect of TGF-β1 on the intracellular XT activity changes in this study.

## Conclusion

The identification of genes that contribute to arthrofibrosis will provide the foundation for the development of antifibrotic therapies. We conclude that AFib retain their profibrotic phenotype upon *in vitro* culture, providing a useful tool to study the cellular mechanisms of arthrofibrotic remodeling. We utilized the myofibroblast marker human XT-I as a molecular readout, knowing that the main regulatory pathways and mediators regulating this enzyme belong to the TGF-β signaling pathway and components thereof. We were able to identify hitherto undescribed potential disease regulator genes, including *TGFB2*, encoding the TGF-β2 isoform and *ITGB8*, which encodes the integrin β8 subunit. In comparison to different fibrotic conditions, our study found a shared and unique gene expression signature, including a novel combination of certain proteoglycan genes in arthrofibrosis, that could be translated to other diseases. This knowledge will be of paramount importance for the future development of therapeutic interventions that are effective in treating or even reversing fibroproliferative diseases.

## Supporting information

**S1 Table. Primer sequences and annealing temperatures ($T_A$) used for qRT-PCR analysis.**
(PDF)

**S1 Fig. No differences in basal and TGF-β1-induced *VCAN* and *SDC2* mRNA expression in AFib and CF.**
(PDF)

## Acknowledgments

We thank Philip Saunders for linguistic advice and Christoph Lichtenberg for technical assistance.

## Author Contributions

**Conceptualization:** Thanh-Diep Ly, Cornelius Knabbe, Isabel Faust-Hinse.

**Data curation:** Thanh-Diep Ly, Meike Sambale, Lara Klösener.

**Formal analysis:** Thanh-Diep Ly.

**Investigation:** Thanh-Diep Ly, Meike Sambale, Lara Klösener, Isabel Faust-Hinse.

**Methodology:** Isabel Faust-Hinse.

**Project administration:** Doris Hendig, Isabel Faust-Hinse.

**Resources:** Philipp Traut, Bastian Fischer, Isabel Faust-Hinse.

**Supervision:** Cornelius Knabbe.

**Validation:** Thanh-Diep Ly, Bastian Fischer, Doris Hendig, Joachim Kuhn.

**Visualization:** Thanh-Diep Ly, Meike Sambale, Lara Klösener.

**Writing – original draft:** Thanh-Diep Ly.

**Writing – review & editing:** Philipp Traut, Bastian Fischer, Doris Hendig, Joachim Kuhn, Cornelius Knabbe, Isabel Faust-Hinse.

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
