## [Decision Letter · Decision Letter 0]

10 Jan 2023

PONE-D-22-27716Understanding of arthrofibrosis: New explorative insights into extracellular matrix remodeling of synovial fibroblastsPLOS ONE

Dear Dr. Faust-Hinse,

Thank you for submitting your manuscript to PLOS ONE. After careful consideration, we feel that it has merit but does not fully meet PLOS ONE’s publication criteria as it currently stands. Therefore, we invite you to submit a revised version of the manuscript that addresses the points raised during the review process.

The reviewers have made a number of comments and suggestions that indicating that the manuscript should be revised and reviewed again before a decision can be recommended on its suitability for publication.

We look forward to receiving your revised manuscript.

Kind regards,

Dominik R. Haudenschild, Ph.D.

Academic Editor

PLOS ONE

Journal Requirements:

Additional Editor Comments:

The reviewers have made a number of comments and suggestions that indicating that the manuscript should be revised and reviewed again before a decision can be recommended on its suitability for publication.

Reviewer 1:

This manuscript describes the remodeling of the extracellular matrix of synovial primary fibroblasts obtained from arthrofibrotic tissues. The study was performed using standard methods on cellular and molecular biology and statistical analysis was used correctly. Limitations of this study are presented in the Discussion and the Conclusions. Indeed, this study opens new issues for a more detailed examination of the kind of GAGs increased after TGF-beta treatment rather than the whole amount of GAGs, the activity of the altered MMPs found in PCR array etc.

Reviewer 2:

The manuscript by Ly et al investigated the differences in cultured fibroblasts isolated from healthy individuals and patients with arthrofibrosis, in terms of xylosyltransferase-1 activity important for collagen crosslinking, collagen and proteoglycan expression, fibrotic genes expression, and cellular contractility, in the presence or absence of TGFb. The manuscript is well-written in a manner that is clear and concise. The major concern, as the authors pointed out, was the lack of sufficient biological replicates and this significantly reduced the scientific rigor of this study and could potentially alter the study’s major conclusion. Although the authors have identified novel genes potentially involved in regulation of arthrofibrosis, the functions and roles of these genes in fibrosis should have been investigated further in this study, as many of the results from this study are not novel and have been published in other areas of fibrosis.

Major points:

1. In Fig 2, based on the XT activity assays, the authors concluded that there was an increase in XT expression or secretion at 96h. However, this could be due to an increase in the enzymatic activity rather than an increase in enzyme expression or secretion, XYLT1 mRNA level should be determined at 96 h. Also, while the author did investigate the basal XYLT1 mRNA level at 2 h and claimed that this time frame is sufficiently short to exclude any residue cytokine effects and epigenetic changes carried over from previous in vivo conditions, they did not cite any supporting reference. Additional experiments aim to detect epigenetic differences in these cells should be included.

2. While there were increased expression in collagens and proteoglycans in fibrotic vs control cells as expected, the results in the presence of TGF were confounding as the study showed that TGF actually inhibited the levels of collagens and proteoglycans in fibrotic cells. This is contradictory to the known effects of TGF as a profibrotic agents. What is the authors’ explanation on this? The dosage of TGF has been shown to be important in determining its biological effects, as too high a dose could have an opposite effect. Due to manufacturer and batch variations in TGFb, have the authors done a titration experiment to determine the optimal dosage for the TGF used in their studies?

Minor points:

1. There is a typo in line 285, “Aecretion”

2. In line 354, “biological replicates” should be technical replicates.

Reviewers' comments:

Reviewer's Responses to Questions

**Comments to the Author**

1. Is the manuscript technically sound, and do the data support the conclusions?

Reviewer #1: Yes

Reviewer #2: No

2. Has the statistical analysis been performed appropriately and rigorously? 

Reviewer #1: Yes

Reviewer #2: Yes

3. Have the authors made all data underlying the findings in their manuscript fully available?

Reviewer #1: Yes

Reviewer #2: Yes

4. Is the manuscript presented in an intelligible fashion and written in standard English?

Reviewer #1: Yes

Reviewer #2: Yes

5. Review Comments to the Author

Reviewer #1: This manuscript describes the remodeling of the extracellular matrix of synovial primary fibroblasts obtained from arthrofibrotic tissues. The study was performed using standard methods on cellular and molecular biology and statistical analysis was used correctly. Limitations of this study are presented in the Discussion and the Conclusions. Indeed, this study opens new issues for a more detailed examination of the kind of GAGs increased after TGF-beta treatment rather than the whole amount of GAGs, the activity of the altered MMPs found in PCR array etc.

Reviewer #2: The manuscript by Ly et al investigated the differences in cultured fibroblasts isolated from healthy individuals and patients with arthrofibrosis, in terms of xylosyltransferase-1 activity important for collagen crosslinking, collagen and proteoglycan expression, fibrotic genes expression, and cellular contractility, in the presence or absence of TGFb. The manuscript is well-written in a manner that is clear and concise. The major concern, as the authors pointed out, was the lack of sufficient biological replicates and this significantly reduced the scientific rigor of this study and could potentially alter the study’s major conclusion. Although the authors have identified novel genes potentially involved in regulation of arthrofibrosis, the functions and roles of these genes in fibrosis should have been investigated further in this study, as many of the results from this study are not novel and have been published in other areas of fibrosis.

Major points:

1. In Fig 2, based on the XT activity assays, the authors concluded that there was an increase in XT expression or secretion at 96h. However, this could be due to an increase in the enzymatic activity rather than an increase in enzyme expression or secretion, XYLT1 mRNA level should be determined at 96 h. Also, while the author did investigate the basal XYLT1 mRNA level at 2 h and claimed that this time frame is sufficiently short to exclude any residue cytokine effects and epigenetic changes carried over from previous in vivo conditions, they did not cite any supporting reference. Additional experiments aim to detect epigenetic differences in these cells should be included.

2. While there were increased expression in collagens and proteoglycans in fibrotic vs control cells as expected, the results in the presence of TGF were confounding as the study showed that TGF actually inhibited the levels of collagens and proteoglycans in fibrotic cells. This is contradictory to the known effects of TGF as a profibrotic agents. What is the authors’ explanation on this? The dosage of TGF has been shown to be important in determining its biological effects, as too high a dose could have an opposite effect. Due to manufacturer and batch variations in TGFb, have the authors done a titration experiment to determine the optimal dosage for the TGF used in their studies?

Minor points:

1. There is a typo in line 285, “Aecretion”

2. In line 354, “biological replicates” should be technical replicates.

6. PLOS authors have the option to publish the peer review history of their article (what does this mean?). If published, this will include your full peer review and any attached files.

Reviewer #1: No

Reviewer #2: No

---

## [Author Response · Author response to Decision Letter 0]

24 Feb 2023

Please see the attachment (Response to Reviewers).

---

## [Decision Letter · Decision Letter 1]

15 May 2023

Understanding of arthrofibrosis: new explorative insights into extracellular matrix remodeling of synovial fibroblasts

PONE-D-22-27716R1

Dear Dr. Faust-Hinse,

We’re pleased to inform you that your manuscript has been judged scientifically suitable for publication and will be formally accepted for publication once it meets all outstanding technical requirements.

Kind regards,

Liangliang Xu

Academic Editor

PLOS ONE

Additional Editor Comments (optional):

Reviewers' comments:

Reviewer's Responses to Questions

**Comments to the Author**

1. If the authors have adequately addressed your comments raised in a previous round of review and you feel that this manuscript is now acceptable for publication, you may indicate that here to bypass the “Comments to the Author” section, enter your conflict of interest statement in the “Confidential to Editor” section, and submit your "Accept" recommendation.

Reviewer #1: All comments have been addressed

Reviewer #2: All comments have been addressed

2. Is the manuscript technically sound, and do the data support the conclusions?

Reviewer #1: Yes

Reviewer #2: Yes

3. Has the statistical analysis been performed appropriately and rigorously? 

Reviewer #1: Yes

Reviewer #2: Yes

4. Have the authors made all data underlying the findings in their manuscript fully available?

Reviewer #1: Yes

Reviewer #2: Yes

5. Is the manuscript presented in an intelligible fashion and written in standard English?

Reviewer #1: Yes

Reviewer #2: Yes

6. Review Comments to the Author

Reviewer #1: After a major revision of the original manuscript, authors answered to the questions mainly by documenting results from the literature. These references demonstrate their choices of the experimental schedule and the interpretation of the results. The manuscript is now ameliorated and complete.

Reviewer #2: (No Response)

7. PLOS authors have the option to publish the peer review history of their article (what does this mean?). If published, this will include your full peer review and any attached files.

Reviewer #1: No

Reviewer #2: **Yes: **Jasper Yik

---

## [Editor Report · Acceptance letter]

17 May 2023

PONE-D-22-27716R1 

Understanding of arthrofibrosis: new explorative insights into extracellular matrix remodeling of synovial fibroblasts 

Dear Dr. Faust-Hinse:

I'm pleased to inform you that your manuscript has been deemed suitable for publication in PLOS ONE. Congratulations! Your manuscript is now with our production department. 

Kind regards, 

on behalf of

Professor Liangliang Xu 

Academic Editor

PLOS ONE